# Research ethics and collaborative research in health and social care: Analysis of UK research ethics policies, scoping review of the literature, and focus group study

Chiara De Poli[1]*, Jan Oyebode[2]

**1** Department of Health Policy and Department of Social Policy, Care Policy and Evaluation Centre, London School of Economics and Political Science, London, United Kingdom, **2** Faculty of Health Studies, Jan Oyebode, Centre for Applied Dementia Studies, University of Bradford, Bradford, United Kingdom

* c.de-poli@lse.ac.uk

**Data Availability Statement:** All relevant data are within the paper and its Supporting Information files.

## Abstract

Current research ethics frameworks were developed on the footprint of biomedical, experimental research and present several pitfalls when applied to non-experimental social sciences. This work explores how the normative principles underpinning policy and regulatory frameworks of research ethics and the related operational processes work in practice in the context of collaborative health and social care research. The work was organised in three phases. First, UK research ethics policy documents were analysed thematically, with themes further organised under the categories of 'Principles' and 'Processes'. Next, we conducted a scoping review of articles about research ethics in the context of collaborative health and social care research, published in English between 2010 and 2022. We then held an exploratory focus group with ten academic researchers with relevant experience to gather their views on how the research ethics system works in practice in England (UK). The thematic framework developed in the first phase supported the analysis of the articles included in the scoping review and of focus group data. The analysis of policy documents identified twelve themes. All were associated to both a principle and a related operational process. The scoping review identified 31 articles. Across these, some themes were barely acknowledged (e.g., Compliance with legislation). Other themes were extensively covered (e.g., The working of Research Ethics Committees), often to discuss issues and limitations in how, in practice, the research ethics system and its processes deal with collaborative research and to suggest options for improvement. Focus group data were largely consistent with the findings of the scoping review. This work provides evidence of the poor alignment between how the research ethics system is normatively expected to work and how it works in practice and offers options that could make research ethics more fit for purpose when addressing collaborative research in health and social care.

**Funding:** The Authors were funded by The Health Foundation, grant number 1274233. The funding source had no involvement in the study design; in the collection, analysis and interpretation of data; in the writing of the articles; and in the decision to submit it for publication.

**Competing interests:** Chiara De Poli had research grant funding from the National Institute for Health and Care Research, School of Social Care Research, grant number 106152/CBF/LSECDP-IF14. She currently has, as co-applicant, research grant funding by the National Institute of Health and Care Research - Three Schools Dementia Programme, grant number 102645/3SDRP/LSEACH-DP05. Jan Oyebode currently has research grant funding from the National Institute for Health Research, grant number 204266, Social Care for People with Young Onset Dementia.

# Introduction

Research ethics and governance represent the regulatory and institutional cornerstones for the conduct of research involving human participants, aiming to oversee its ethical quality and protect research participants from harmful research practices. Since the Nuremberg code (1947) [1], respect for autonomy, justice, beneficence (i.e., to do good), and non-maleficence (i.e., to do no harm) have been recognised as core ethical principles of biomedical, experimental research involving human participants. These principles were embedded in international guidance and regulation (e.g., the Declaration of Helsinki by the World Medical Association (1964) [2,3], the International Ethical Guidelines for Biomedical Research Involving Human Subjects of the Council for International Organizations of Medical Sciences (CIOMS, 2016) [4] and in national research ethics frameworks (e.g., the US Belmont Report (1978) [5], the Canadian Tri-Council Policy Statement [6], and the UK Policy Framework for Health and Social Care Research [7]). Over time the application of this research ethics framework has been expanded from regulation of traditional biomedical research to also regulate social sciences, including health and social care research.

Extant research on research ethics has underscored the intrinsic complexity of establishing whether ethics oversight results in ethical research and participant protection. The use of processes and structures as surrogate measures for effectiveness of research ethics has been found particularly unsatisfactory [8–10]. Although there are several reasons to believe that research ethics processes and structures contribute to ensuring ethical research and the protection of research participants, little empirical evidence is available on whether and how research ethics institutions actually achieve these ends [9,11,12]. Alongside assumed benefits, negative or unwanted consequences of the research ethics system have been documented. At the procedural level, the apparent arbitrary nature of decision-making, long delays in obtaining research approvals, and the bureaucratic restrictions imposed on the conduct of studies are a frequent source of problems [8,13]. At the organisational level, the establishment of a research ethics industry [14], with a perceived emphasis on box ticking [15,16] and rule fetishization [17], ultimately concerned with issues around risk, litigation, and institutional reputation, seems to have contributed to an 'ethics creep' [17]. At an epistemological level, the problems of using a framework rooted in experimental, biomedical research to assess social sciences and research using non-experimental methods (e.g., big data research [18], social media research [19], research using machine learning [20], ethnography [21] and digital ethnography [22]) are well documented [23–26].

Research using qualitative or mixed methods [27] and less codified and predictable designs, such as participatory research (e.g., participatory action research, community-based participatory research) [28] and research using collaborative approaches (e.g., co-creation, co-design, co-production) [29], have been particularly affected. As a consequence of the fundamental poor alignment between the biomedical framework of research ethics and qualitative, participatory, and collaborative research, researchers have described their experience of navigating the research ethics system as "jumping through hoops" or "walking a tightrope", or "something to get through" [30,31]. The requirement to submit a detailed research protocol to prospectively outline research activities clashes with the emergent nature of qualitative, participatory, and collaborative research [29,30]. The bureaucratic practices around consent that are used to operationalise the principle of autonomy are often unworkable for research with a strong relational component [17,32–34]. Additionally, research participants and co-researchers may disagree with the way the principle of participant protection is interpreted and risks and benefits are assessed by research ethics institutions [29,35,36]. These may be perceived as leaning towards paternalism, in particular in the case of populations with characteristics that are perceived as making them vulnerable [37–39].

If some of these features of qualitative, participatory, and collaborative research result in great scrutiny by research ethics institutions, other features with equally relevant ethical implications are not given due attention. For example, current research ethics frameworks designed to address the principle of justice (which requires the equitable distribution of both the burdens and the benefits of participation in research [40]) fail to consider how power relations between individuals or groups shape research and do not offer any mechanisms to help address the power differentials intrinsic to research [41]. Research ethics processes also have a blind spot in relation to gatekeepers, who have a substantial influence on who gets to participate and, conversely, who is excluded from research. There is no mechanism in place to oversee how they perform their role and whether they impact upon free choice of potential participants to take part in studies [34].

Alongside these issues reported for qualitative, participatory, and collaborative research that reaches the stage of seeking ethics approval, the academic debate has also aired concerns that research which is anticipated to be met with resistance by research ethics institutions is not actually pursued [17,42,43]. The perceived or expected barriers might deter researchers from conducting research on sensitive topics, involving vulnerable groups, or using more innovative methods. At best, this could contribute to homogenisation of the research landscape. At worst, it could undermine the role of qualitative, participatory, and collaborative research in promoting research inclusivity and social justice, and in answering research questions that no other research methods could address.

In order to move these debates forward, we conducted a study on research ethics aimed at generating a set of practical recommendations for improving how the research ethics system deals with participatory and collaborative research approaches (collaborative only, hereon). In this context, we used collaborative research as an umbrella term for various research approaches (e.g., participatory action research, community-based participatory research, co-creation, co-design, co-production) where participants are actively involved in shaping the research, beyond simply providing data, and where the primacy of academic knowledge is challenged by other types of knowledge (e.g., based on lived experience).

The study was organised in five consecutive phases. In the first phase, we carried out an analysis of UK research ethics policies, which informed a scoping review of the literature (phase 2). We then held an exploratory focus group with academic researchers in this field to understand their perspectives on the topic (phase 3). Results of these three phases informed a two-round Delphi study, involving academic researchers with experience of conducting participatory and collaborative research involving vulnerable groups in England (UK). The Delphi study aimed to generate consensus on what changes to the research ethics system should be considered to improve the ethics oversight of collaborative research (phase 4). A final focus group with experts was organised to inform the practical recommendations and explore their expected benefit (phase 5). This article reports the results of the first three phases of the work, whilst the remaining phases are reported elsewhere [44].

Through the analysis of policies (phase 1), we aimed to (i) understand what the UK research ethics system is intended to achieve and how it is designed to work, (ii) identify its underpinning principles, and (iii) map the operational processes and procedures which are designed and implemented to achieve the principles.

Since our work was carried out in the context of a UK-based study, the focus of the analysis was limited to UK research ethics policies, which reflect relevant domestic legislation (as set out in Appendix 2 of the UK Policy Framework for Health and Social Care Research [7],), but also draw on international standards, governance mechanisms, and good research practice (as per [7], para 3.4). By design, such policies have a deliberate wide scope: they do not, and possibly cannot, exhaustively compile principles, requirements, and standards that may be relevant

for specific types of research, which are left to organisations with responsibilities under the national policy framework. In this sense, the UK system largely shares the bedrock of research ethics principles with other countries [45–48] and can be considered an example of a modern research ethics system. The analysis of policies was also instrumental to the development of the analytical framework of 'Principles' and 'Processes' that supported the scoping review of the literature that followed (phase 2).

The aim of the review was two-fold. Firstly, we aimed to understand how the normative principles and operational processes of research ethics play out in actual research that adopts participatory or collaborative approaches, often using qualitative methods, in the health and social care field. Secondly, we set out to map the recommendations that the literature had suggested to improve how the research ethics system deals with this type of research. The scoping review approach was deemed fit for the purpose of efficiently gathering and examining the extent, range, and nature of the literature available on this topic.

In phase 3, the exploratory focus group aimed to bring to light actual experiences of navigating the research ethics system from the perspective of active academic researchers with experience of conducting collaborative research in England (UK). By gathering their experiences, we were able to identify current patterns in the English context and to read them against the background of the literature review.

## Methods

### Analysis of UK research ethics policies

The analysis focused on the UK Policy Framework for Health and Social Care Research [7] and on UK Health Research Authority (HRA) policies publicly available online [49,50]. Policy documents were analysed thematically. The analysis proceeded deductively at first, with themes identified from the research ethics principles stated in the UK Policy Framework for Health and Social Care Research. The initial codebook was then expanded inductively, to include additional themes that were not codified as principles in the UK Policy Framework, but that appeared relevant. To help systematise the data collected, we divided the material under each theme into two categories: 'Principles' for data that referred to underpinning principles of the UK research ethics system, and 'Processes' for data describing the operational processes and procedures supporting the implementation of the principle (Table 1). The Authors worked collaboratively: CDP started the data extraction and discussed the emerging results with JO as the analysis progressed. Instances of uncertainty or ambiguity were resolved through ongoing discussion.

### Scoping review of the literature

The scoping review was conducted following the Arksey and O'Malley framework [51] and is reported based on the PRISMA extension for scoping reviews (PRISMA-ScR) [52]. The review was guided by the following questions: How does the current research ethics system work for collaborative research in health and social care? What are the challenges that the current

**Table 1. Thematic framework underpinning the analysis of UK research ethics policies.**

| Theme X | Current system, as per policy documents |
|---|---|
| 1. Principles of the research ethics system (Principles) | |
| 2. Operationalization/Implementation of the research ethics principles (Processes) | |

**Table 2. Thematic framework underpinning the scoping review of the literature and the analysis of focus group data.**

| Theme X | Current system | Options for improvement |
|---|---|---|
| Principles of the research ethics system (Principles) | | |
| Operationalization/Implementation of the research ethics principles (Processes) | | |

research ethics framework poses to collaborative research in health and social care? What options have been discussed in the literature to overcome these challenges?

**Eligibility criteria.** Articles examining the research ethics and governance systems in the context of collaborative research in the field of health and social care were included. They had to be written in English and published in peer-reviewed journals in the period January 2010-May 2020. The original search was updated in December 2022, to identify articles published between June 2020-December 2022. Three online databases (Web of Science, PubMed/Medline, PsycInfo) were searched in June 2020, using search strings available in S1 File. In December 2022, we re-ran the original search using the same online databases and the same search criteria. Retrieved studies were imported into a reference management software. After removing duplicates, titles, and abstracts of the retrieved results were screened by CDP for eligibility against the inclusion criteria. Full text of all possible eligible articles was retrieved and screened by CDP, with JO screening the articles for which eligibility was uncertain.

**Data extraction and analysis.** Data extraction and analysis was conducted by CDP and iteratively discussed and reviewed by JO. The analysis was supported by an analytic framework organised around the themes identified in the analysis of policy documents. At the level of each theme, the analytical framework was organised as a 2X2 matrix. On the first dimension we placed the categories of 'Principles' and 'Processes', consistently with the way we had analysed the policy documents. On the second dimension, we used the categories of 'Current system' and 'Options for improvement' (Table 2), to map the debate in relation to how the current research systems deal with collaborative research and to collect suggestions put forward to ensure a better fit of research ethics in the context of collaborative research. The resulting analytic framework was applied to each article. Each article was also charted in relation to its typology, country in which the research was conducted, research approach, and research population(s).

As recommended by methodological guidance on scoping reviews [51,53], this analytical process allowed us to describe the literature available in this field. It also allowed us to systematically document how the set of a priori themes, derived from the policy documents, had been discussed in the literature. Lastly, it enabled us to identify, in relation to each theme, how the system currently works and opportunities for improvement.

**Exploratory focus group.** Following approval by the London School of Economics and Political Science, as per the School's research ethics policy, the exploratory focus group was conducted online, via Zoom, in May 2020 to discuss actual experiences of navigating the English research ethics system in the context collaborative research in the health and social care field. The emphasis in recruitment of participants was on identifying active researchers with relevant experience of undertaking collaborative research in health and social care, and therefore of having applied for ethical review in England (UK).

Twelve participants were identified or snowballed via the professional networks of the research team and invited by e-mail to take part. Ten academic researchers with different levels of relevant experience, some undertaking research involving populations deemed vulnerable, expressed interest and were sent background information about the study and what their participation would entail. All gave written consent to take part in the study (Table 3).

**Table 3. Participants in the exploratory focus group.**

| Participant id | Gender | Years of experience | Research interest |
|---|---|---|---|
| FG1-1 | female | 5–10 years | dementia care research |
| FG1-2 | female | 5–10 years | care homes research |
| FG1-3 | female | 5–10 years | dementia care research |
| FG1-4 | female | 0–5 years | dementia care research |
| FG1-5 | male | more than 10 years | health services research |
| FG1-6 | female | more than 10 years | patient and public involvement |
| FG1-7 | male | more than 10 years | social care research |
| FG1-8 | male | more than 10 years | social care research |
| FG1-9 | female | 0–5 years | dementia care research |
| FG1-10 | female | 5–10 years | palliative care research |

The group was facilitated by JO with assistance from CDP, both with similar research interests and experience. The discussion was audio-recorded with consent from participants.

The discussion lasted about two hours and was guided by two broad questions. We started off by asking participants about issues they had encountered in obtaining research governance and ethics approval for studies using collaborative approaches and involving groups deemed vulnerable. The second question invited participants to consider what changes could ensure that research governance and ethics approval processes were better suited for collaborative research with vulnerable groups.

The focus group was transcribed verbatim. Data were imported to Nvivo12 and analysed thematically using a deductive approach supported by the analytical framework developed for the study [54] (Table 2).

## Results

### Analysis of UK research ethics policies

The analysis of the UK Policy Framework [7] and HRA documents [49,50] resulted in 12 themes (Table 4).

Theme 1—General ethical principles is a general theme which reflected Principle 3 (Scientific and Ethical Conduct Research) of the UK Policy Framework, by which research projects are expected to be scientifically sound and guided by ethical principles in all their aspects [7].

We identified both principles and processes for three of the 11 remaining themes: Theme 3—Protection of research participants corresponds to Principle 8 (Benefits and risks) of the UK Policy Framework. The corresponding operational process revolves around the role of Research Ethics Committees (RECs, known as Institutional Review Boards (IRBs) in the US and Canada) to ensure that the rights, safety, dignity, and wellbeing of research participants are adequately protected [49]. Theme 6—The working of RECs reflects Principle 9 (Approval) of the UK Policy Framework. Operationally, this principle relies on Research Ethics Committees (RECs) providing ethical review of new applications and keeping approved applications under review [49]. The HRA Standard Operating Procedures for RECs make specific provision regarding the review process (e.g., 'flagged' RECs, Proportionate Review Service, expedited review) for different types of research [50]. Theme 7—The research protocol is equivalent to the Principle 6 (Protocol) of the UK Policy Framework. From an operational perspective, this translates into a requirement to submit the standard protocol for any new research study and a standard Notice of Substantial Amendment when significant changes to the original study are proposed [50].

**Table 4. Themes, principles, and processes of the research ethics system identified from the analysis of UK research ethics policies.**

| Theme | Principle | Process |
|---|---|---|
| 1—General ethical principles | Principle 3—Scientific and Ethical Conduct Research. Research projects are scientifically sound and guided by ethical principles in all their aspects [7] | |
| 2—Patient, service user, and public involvement | Principle 4—Patient, service user and public involvement. Patients, service users and the public are involved in the design, management, conduct and dissemination of research, unless otherwise justified [7] | |
| 3—Protection of research participants | Principle 8 –Benefits and risks. The safety and well-being of the individual prevail over the interests of science and society. Before the research project is started, any anticipated benefit for the individual participant and other present and future recipients of the health or social care in question is weighed against the foreseeable risks and inconveniences once they have been mitigated [7] | The Research Ethics Service (RES) has a duty to provide an efficient and robust ethics review service that maximises UK competitiveness for health research and maximises the return from investment in the UK, while protecting participants and researchers [49] Research Ethics Committees protect the rights, safety, dignity and wellbeing of research participants [7] |
| 4—Privacy and confidentiality | Principle 14 –Respect for privacy. All information collected for or as part of the research project is recorded, handled, and stored appropriately and in such a way and for such time that it can be accurately reported, interpreted, and verified, while the confidentiality of individual research participants remains appropriately protected. Data and tissue collections are managed in a transparent way that demonstrates commitment to their appropriate use for research and appropriate protection of privacy [7] | |
| 5—Role and competence of researchers | Principle 2—Competence. All the people involved in managing and conducting a research project are qualified by education, training and experience, or otherwise competent under the supervision of a suitably qualified person, to perform their tasks [7] | |
| 6—The working of RECs | Principle 9—Approval. A research project is started only if a research ethics committee and any other relevant approval body have favourably reviewed the research proposal or protocol and related information, where their review is expected or required [7] | The RES aims to prove robust, proportionate and responsive ethical review of research through Research Ethics Committees (RECs) [49] REC should keep under review the favourable ethical opinion given to any research study in the light of regular progress reports and significant developments in the research [50] Depending on the type of research, applications may be reviewed by a 'flagged' REC, i.e. RECs designated for review of particular types of application due to having relevant professional, academic and ethical expertise among the Committee's membership [50] Research studies raising no material ethical issues, including projects involving straightforward issues which can be identified and managed routinely in accordance with standard research practice and existing guidelines, can be assessed under the Proportionate Review Service (PRS) regime [50] RECs must always adopt a proportionate approach in assessing whether a non substantial amendment may require a new application. A new application should only be required where a proposed amendment would **fundamentally**\* alter the nature of the research and the extent of the involvement of, or risk to, existing and/or potential participants [50] Under specific circumstances the RES can consider a research study for expedited review [50] |
| 7—The research protocol | Principle 6—Protocol. The design and procedure of the research are clearly described and justified in a research proposal or protocol, where applicable conforming to a standard template and/or specified contents [7] | All new applications for ethical review to a Research Ethics Committee (REC) in the UK should be submitted on the standard on-line REC application form in the Integrated Research Application System (IRAS) [7,50] Substantial amendments must be requested when the proposed changes to the original study will affect the research "to a significant degree". Particular account should be taken of any implications for the safety or welfare of participants, and of any information that participants might require to give informed consent to continue to participate in the research as amended. It is recommended that where there is any doubt about the potential implications of the amendment for participants, it should be treated as a substantial amendment and reviewed by the REC. The RES Notice of Substantial Amendment form should be used and submitted to the REC electronically together with the documents that have been modified [50]. |

*(Continued)*

**Table 4.** (Continued)

| Theme | Principle | Process |
|---|---|---|
| 8—Seeking consent | Principle 12—Choice. Research participants (Either directly, or indirectly through the involvement of data or tissue that could identify them) are afforded respect and autonomy, taking account of their capacity to understand. Where there is a difference between the research and the standard practice that they might otherwise experience, research participants are given information to understand the distinction and make a choice, unless a research ethics committee agrees otherwise. Where participants' explicit consent is sought, it is voluntary and informed. Where consent is refused or withdrawn, this is done without reprisal [7] | |
| 9—Compliance with legislation | Principle 7—Legality. The researchers and sponsor familiarise themselves with relevant legislation and guidance in respect of managing and conducting the research [7] | |
| 10—Integrity, quality, and transparency of research | Principle 5—Integrity, Quality, and Transparency Research. Research is designed, reviewed, managed and undertaken in a way that ensures integrity, quality, and transparency [7] | |
| 11—Accessible findings | Principle 11—Accessible Finding. The findings, whether positive or negative, are made accessible, with adequate consent and privacy safeguards, in a timely manner after they have finished, in compliance with any applicable regulatory standards, i.e., legal requirements or expectations of regulators. In addition, where appropriate, information about the findings of the research is available, in a suitable format and timely manner, to those who took part in it, unless otherwise justified [7] | |
| 12—Benefits from research | The HRA has the mission to facilitate and promote ethical research that is of potential benefit to participants, science and society [49] | |

*Bold and underlined in the original document.

The remaining eight themes substantially mirrored principles of the UK Policy Framework, but no corresponding operational process could be identified.

## Scoping review of the literature

The search conducted in June 2020 yielded 230 references, of which 166 were unique records. After screening titles and abstracts, 109 articles were excluded. The full texts of the remaining 57 papers were retrieved and reviewed. A total of 30 articles were deemed relevant for inclusion. The December 2022 update search identified 23 further references, of which 10 were unique records. After screening titles and abstracts, eight articles were excluded. The full texts of the two remaining papers were retrieved and assessed, and one was included. Combining the two searches, a total of 31 articles were deemed relevant for inclusion in the review (Table 5). The screening process is illustrated in Fig 1.

Seven articles reported research based in Canada [28,30,56,61–63,75], seven reported research based in the USA [38,57,60,67,68,73,77], four in England or the UK [35,55,76,78], two in Scandinavian countries [32,71], one in Malta [69], and one in Australia [64]. Four referred to multiple countries (e.g., low and middle income countries [72], Ireland and the UK [59]). Of these, two did not specify the geographical setting in full [34,58]. Five articles did not provide any reference to the geographical setting of their work [29,65,66,70,74].

**Table 5.** Overview of characteristics of the articles included in the scoping review of the literature.

| Reference | Type of article | Country | Type of research | Type of population |
|---|---|---|---|---|
| Burns et al (2014) [55] | Case study, with critical analysis of research ethics issues | UK | Participatory organizational research Co-production | Elderly people |
| Chabot et al (2012) [56] | Case study, with critical analysis of research ethics issues | Canada | Participatory Action Research | Young people, aged 16–24 |
| Cross et al (2015) [57] | Methodological article on CBPR, and ethics challenges | USA | Community-based participatory research (CBPR) | Not specified |
| Damianakis et al (2012) [58] | Two case studies, with critical analysis of research ethics issues | Case 1: NR Case 2: Canada | Case 1: arts-based social work Case 2: participatory policy making | "Small, connected community" |
| Doyle et al (2017) [59] | Synthesis of literature on research ethics, with framework to aid research ethics approval process | UK and Ireland | Qualitative research, Participatory Action Research | Not specified |
| Fiscella et al (2015) [60] | Commentary, critical reflection on research ethics issues | USA | Quality Improvement Research (QIR) vs Quality Improvement (QI) | Not specified |
| Goodyear-Smith et al (2015) [29] | Commentary, critical reflection on research ethics issues | NR | Co-design and implementation research | Not specified |
| Gustafson et al (2014) [61] | Case study, with critical analysis of research ethics issues | Canada | Participatory Action Research | People with disabilities |
| Guta et al (2010) [62] | Content analysis of REB ethics review documentation | Canada | Community-based participatory research (CBPR) | Not specified |
| Guta et al (2012) [28] | Qualitative study—interviews with REB/IRB members, staff, and other informants | Canada | Community-based participatory research (CBPR) | Not specified |
| Guta et al (2013) [63] | Qualitative study—interviews with REB/IRB members, staff, and other informants | Canada | Community-based participatory research (CBPR) | Not specified |
| Iedema et al (2013) [64] | Two case studies, with critical reflection on research ethics issues | Australia | Qualitative research | Case 1: staff and patients involved in clinical incidents Case 2: patients and families who had been involved in clinical incidents |
| Lange et al (2013) [65] | Theoretical article on vulnerability in research, complemented by two empirical case studies | NR | Not specified | Vulnerable groups broadly defined |
| Lavery (2018) [66] | Commentary, critical reflection on research ethics issues | NR | Community-engaged research (CEnR) | People living with HIV/AIDS |
| McCormack et al (2012) [30] | Commentary, critical reflection on research ethics issues | Canada | Qualitative research | Not specified |
| McDonald et al (2021) [67] | Two case studies, with critical reflection on research ethics issues | USA | Community- Based Participatory Research, photovoice | Case 1: grandparent caregivers Case 2: Lesbian, Gay, Bisexual, Transgender, Queer (LGBTQ) former foster youth |
| Noorani et al (2017) [35] | Two case studies, with critical reflection on research ethics issues | England (UK) | Participatory research | Individuals with mental health problems |
| Opsal et al (2016) [68] | commentary, critical reflection on research ethics issues | USA | Qualitative research | Vulnerable groups broadly defined |
| Øye et al (2016) [34] | Two case studies, with critical analysis of research ethics issues | Multiple Case 1: Denmark Case 2 and 3: NR | Qualitative research | Vulnerable groups broadly defined |
| Øye et al (2019) [32] | Two case studies, with critical reflection on research ethics issues | Norway, Denmark | Collaborative research | Not specified |
| Petrova et al (2016) [69] | Case study, with critical reflection on research ethics issues | Malta | Qualitative research | Practice development nurses (PDNs) |

(*Continued*)

**Table 5.** (Continued)

| Reference | Type of article | Country | Type of research | Type of population |
|---|---|---|---|---|
| Ponterotto (2013) [70] | Methodological article on PAR involving people experiencing mental health problems | NR | Qualitative research, Participatory Action Research | Individuals with mental health problems |
| Rink et al (2013) [71] | Case study, with critical reflection on research ethics issues | Greenland | Community-based participatory research (CBPR) | Remote community |
| Ross et al (2010) [38] | Theoretical article—research ethics framework for CEnR | USA | Community-engaged research (CEnR) | Not specified |
| Ruiz-Casares (2014) [72] | Commentary, critical reflection on research ethics issues | LMICs | Participatory research approaches | Individuals with mental health problems |
| Shore et al (2011) [73] | Survey of community-engaged research studies reviewed by Research Ethics Boards | USA | Community-engaged research (CEnR) | Not specified |
| Tamariz et al (2015) [74] | Systematic review of barriers and facilitators of CBPR ethics oversight | NR | Community-based participatory research (CBPR) | Not specified |
| Townsend et al (2010) [75] | Case study, with critical reflection on research ethics issues | Canada | Qualitative research | Individuals living with Rheumatoid Arthritis |
| Whiting et al (2010) [76] | Case study, with critical reflection on research ethics issues | England (UK) | Qualitative research | Palliative care patients |
| Wolf et al (2010) [77] | Case study, with critical reflection on research ethics issues | USA | Community-based participatory research (CBPR) | Deprived community |
| Yanar et al (2016) [78] | Case study, with critical reflection on research ethics issues | England (UK) | Participatory Action Research | Young people |

In terms of research design, 15 articles adopted a single or multiple case study design [32,34,71,75–78,35,55,56,58,61,64,67,69]. The majority of these presented recommendations for policy and research practice derived from the authors' experiences of research ethics-related issues they had faced and navigated in their own studies. Six articles were commentaries offering critical reflections on research ethics, without reference to a specific empirical study [29,30,60,66,68,72]. Four articles reported on empirical studies that generated primary data: Guta et al's work discussed the results of a content analysis of ethics review documentation submitted to Research Ethics Boards (REBs) [62], complemented by a qualitative study based on interviews with REC members, staff, and other key informants [28,63], whereas Shore et al's article reported the results of a survey of community-engaged research studies reviewed by RECs [73].

Of the remaining articles, two were methodological and discussed research ethics implications of Community-based participatory research [57] and Participatory Action Research [67]. A further two were theoretical in nature: Ross et al presented a research ethics framework to support Community Engaged Research [38]; Lange et al [65] discussed the concept of vulnerability and its implications for research ethics. The remaining two were literature reviews. Doyle et al provided a synthesis of the literature that supported the design of a framework to aid the research ethics approval process by RECs [59]. Tamariz and colleagues' systematic review identified the most common perceived barriers and facilitators to evaluating research ethics oversight for Community based participatory research [74].

Research methodologies underpinning the reviewed articles were wide-ranging. Some articles referred to participatory or collaborative research [32,35,72] or community engaged research [38,66,73] as loosely defined umbrella terms, or emphasised the use of qualitative research methods [30,34,59,64,68–70,75,76]. Other articles were grounded in specific definitions and epistemological traditions, such as Community-based participatory research [28,57,62,63,67,71,74,77] or Participatory Action Research [56,61,78]. One article positioned co-design as a collaborative approach to implementation research [29], one focused on

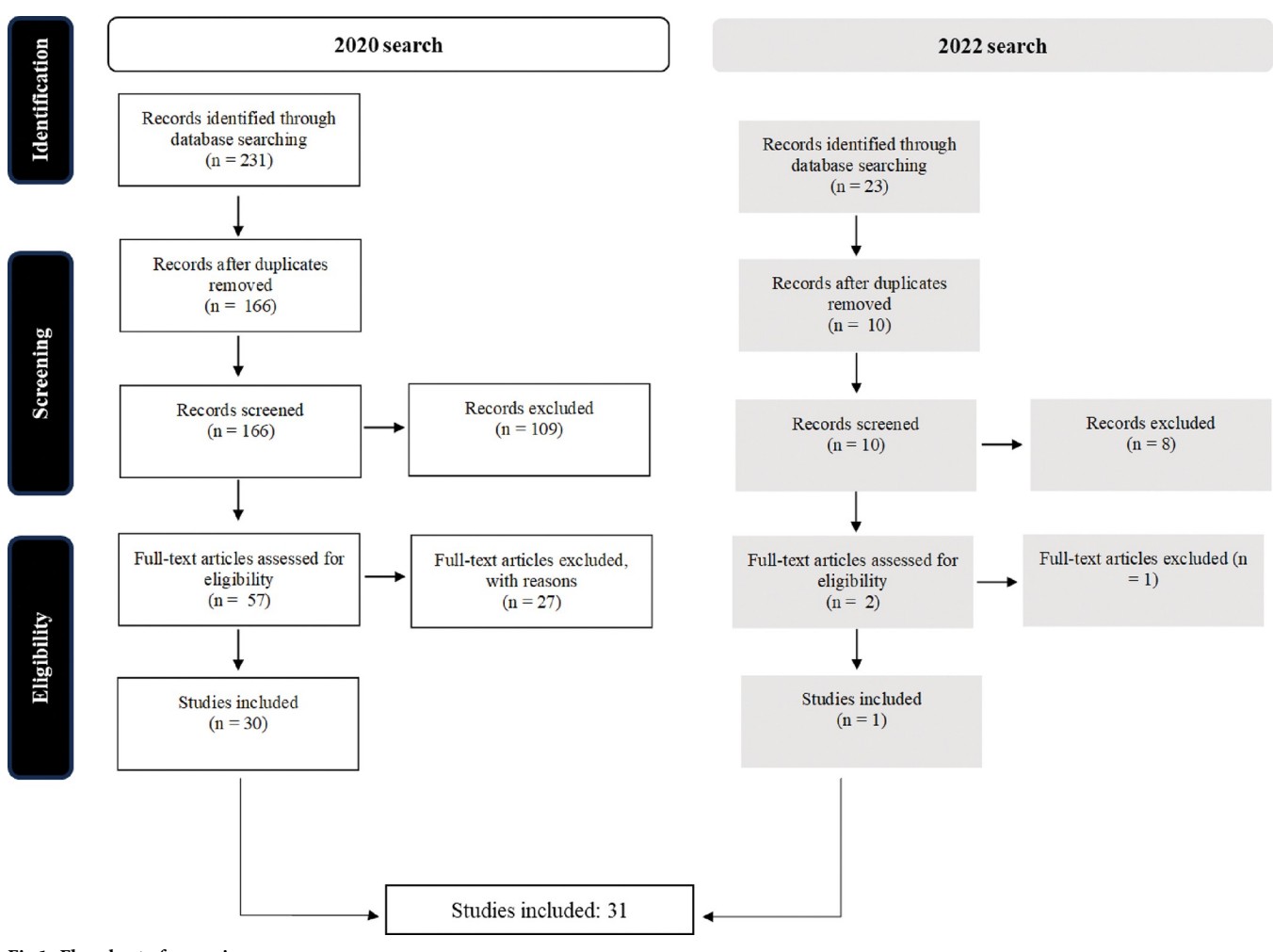

**Fig 1. Flowchart of screening process.**

collaborative approaches in quality improvement [60], and one referred to co-production in the context of participatory organisational research [55].

Study populations could be classified into two broad groups. About a third of the articles did not describe or provide a definition of their study population [28,29,74,30,34,38,57,60,62,63,73]. Such articles typically focused on a specific methodological approach and discussed its implications for research ethics, regardless of the study population or research setting. Of the remaining studies, three focused on vulnerable individuals broadly defined [34,65,68] and the others focused on specific populations deemed vulnerable. In these articles, vulnerability was defined according to demographic characteristics such as age (e.g., young [56,78] or older people [55]), illness (e.g., mental health problems [35,70,72], palliative care patients [76], people living with HIV/AIDS [66]) or disability [61], socio-economic deprivation [77], geography [71]. In other cases, vulnerability was framed as situational, being defined by individual circumstances at a specific point in time. These included grandparent caregivers [67], Lesbian, Gay, Bisexual, Transgender, Queer (LGBTQ) former foster youth [67], participants from small connected communities, which may pose challenges to the anonymity and confidentiality of research participants [58,69], or people diagnosed with a long-term condition who may move through vulnerable moments in their illness trajectory (such as

rheumatoid arthritis [75]), and clinical staff and patients who had been involved in clinical incidents which made them temporarily vulnerable [64].

We found wide variation in the level of coverage of the twelve analytical themes (Table 6). A few themes have been sparsely discussed. The theme on compliance with legislation was touched on only by one article. This highlighted the possible tension in decision-making processes about research ethics when pieces of relevant legislation or regulation are broad in scope and require (or allow) some degree of interpretation [77]. The theme on integrity, quality and transparency of research was discussed only by Øye and colleagues in two different but related articles. In the first, they discussed how gatekeepers may influence the way in which the recruitment of participants is carried out and reported [34]. In the second, they questioned the principle of scientific integrity and research independence in the context of collaborative research, when stakeholder groups with different interests and agendas, necessarily negotiate research plans and the reporting of research results [32]. The theme of accessible findings was discussed in two articles, suggesting ways in which RECs could support wider dissemination of research, beyond academic circles [57,62].

Other themes have been explored more widely, often using the experiences, positive or negative, of their authors as a starting point to discuss how the ethics system currently handles collaborative research, to highlight good practices and identify recommendations for improvement. Several articles highlighted that the biomedical and technocratic approach of research ethics processes and the way in which RECs work are poorly aligned to collaborative research, its relational nature, emergent designs, inclusive approach, and do not allow the flexibility and ongoing adaptations that such research requires [28,29].

The literature has discussed the tension between the involvement of patients, service users, and the public in research–a key feature of collaborative research–and the protection of research participants–a primary concern of research ethics. Although the need for appropriate participant protection was consistently recognised [34,75,76], some authors highlighted that, at times, RECs showed an over-protective attitude towards participants, which ultimately could affect agency of individuals and their participation and inclusion in research [38,61,65,68,72]. Therefore, they argued for an overhaul of the way risks and benefits associated with qualitative and collaborative research are framed and assessed, considering that risks are often low (and lower than for biomedical research), whilst benefits from participation are apparent and should not be dismissed [38,57,59,60,73,76,78]. Moreover, they highlighted that, given the nature of collaborative research, benefits and risks can unfold both at the individual and at the community level and should be assessed accordingly [29].

Consent-seeking practices can be seen as an example of how research ethics principles are operationalised in the context of collaborative research [35]. The suitability and appropriateness of traditional consent-seeking procedures, by which participants are requested to give one-off consent in writing, were questioned with specific reference to collaborative research [55,57]. Similarly, traditional research ethics systems were perceived as being uneasy about co-researchers identified from groups labelled or potentially perceived as vulnerable [32,67,78].

The role of the researcher and the nature of the research protocol were also recurring themes. Some authors contended that the role of the researcher is relational and constructed continuously on the spot [34]. They argued that researcher's skills and experience in attending to ethical relationships with participants and in self-reflection should be integrated into research ethics frameworks [72,75] and considered when defining acceptable levels of risk tolerance [59].

Several authors commented on the constraints of the biomedical framework that underpins the research protocols required by RECs [62,74]. Standard protocols do not fully allow articulation of the relational nature of collaborative research (e.g., to define the nature of the

**Table 6. Results of the scoping review of the literature, by theme.**

| Theme | Current system | Options for improvement |
|---|---|---|
| 1—General ethical principles | Principles<br>Processes<br>• Poor alignment of research ethics approval process with collaborative research [57]<br>• Tension between the time that the ethical approval process takes and keeping momentum in collaborative research studies [71] | Principles<br>• Need to adopt a more participatory, flexible research ethics framework, more attuned to the perspective of the communities and research participants [57,62,74]<br>• Need for a more nuanced research ethics oversight system involving higher levels of tolerance for some research and providing a rationale for contextual sensitivity [59]<br>• Need to rebalance the principles of vulnerability and empowerment [61]<br>• Need for a reflexive approach throughout the research process [75]<br>• Adopting the concept of situated ethics, i.e. ethics which is conditional on the situation [34]<br>Processes<br>• Call for a move towards more openness and collegiality in the review process and more regular opportunities for dialogue between researchers and RECs [30]<br>• Need for a time-effective ethics review process [71]<br>• Considering a review and monitoring process, upon completion of a study [73]<br>• Providing researchers with space to justify deviating from standard procedures and to request accommodations or waivers of normal requirements [62] |
| 2—Patient, service user, and public involvement | Principles<br>• Existing practices of designing research independent of the community of interest and seeking approval from RECs before interacting with potential participants violate ethical principles when considered in relationship to communities [57]<br>• Collaborative research blur the differences between researchers and research participants, undermining the principle of autonomy and voluntary participation [32]<br>• Concerns around the consequences that a biomedical framework pivoted around individualized vulnerability and binarized mental capacity may have on participation and on privileging/depriviliging of voices [35]<br>Processes<br>• Questioning of the role of gatekeepers [34,75,76]<br>• Principle of anonymity can undermine the role of co-researchers, when they want to be publicly recognised for their participation [78] | Principles<br>• Need to rethink how to identify individual or organisation with a *legitimate* interest in the conduct or outcomes of the research [66,74,77]<br>• Need to develop an account of the duties involved in responding to vulnerability that avoids stereotyping and paternalism and is consistent with the principle of respect for individual autonomy [65]<br>Processes• Need to qualify the position of expert advisors [55,56] |
| 3—Protection of research participants | Principles• Rules for ethical research as operationalized by RECs as 'rules of policy' designed to protect research institutions from risk vs ethical 'first principles' designed to protect research volunteers from potential harm [57]• Individual-based ethical frameworks (aiming to protect the autonomy and rights of individuals participants) are not necessarily of benefit to the community at large, who are collectively involved in the research process [29]• Power imbalance created and reproduced by dominant research ethics frameworks, with researchers constructed as responsible for managing the vulnerability of the research population and research subjects constructed as dependent on researchers for their protection and safe journey through the process [61]• Tension between risks and benefits associated with research, e.g., risks framed at the level of individual participants vs benefits framed at the community level [73,74]<br>• Protectionist attitudes of RECs towards *individuals* with marginalised identities or sensitive backgrounds [61,68,72]<br>• Legal/risk-based framework that RECs use in their assessment of risks vs benefits associated with participation in research [68]<br>Processes• Use of procedures unsuitable for collaborative research which may put communities at greater risk, wasting resources, and further marginalizing vulnerable communities rather than contributing to generating the benefits expected by collaborative research [57,65,68] | Principles<br>• Reciprocity of collaborative approaches to diminish participant risk<br>• Power-sharing to reduce inequality and empower vulnerable communities [29,72]<br>• Framing vulnerability in context [61]<br>• Reframing risk vs benefit assessment [57], assessing levels of participant protection/risk tolerance vs type of research (biomedical vs health services research) [59,60]<br>• Reframing concept of research participant (the community within which research is carried out, in addition to individuals taking part in research) [57,74,79]<br>Processes<br>• RECs need to give credit to measures for power-sharing in collaborative research processes and researchers should make such measures explicit in their ethics applications [29]<br>• Collaborative research requires a comprehensive proportionate benefit vs risk assessment that addresses [38]<br>• Use of anonymity should be agreed on with research participants when it is needed to afford them protection [78] |

*(Continued)*

**Table 6.** (Continued)

| Theme | Current system | Options for improvement |
|---|---|---|
| 4—Privacy and confidentiality | Principles• Questioning of the research ethical principle of anonymity to protect the informants' confidentiality and privacy of participants of collaborative research [34], including small connected communities [58,69]• Understanding of privacy is context-specific, existing regulations may not find easy fit in different settings [72]Processes• Dispute of the compulsory use of anonymity/anonymisation for research participants [59,78] | Principles<br> • Privacy, confidentiality and anonymity should be negotiated with participants [30,34,78]<br> • Privacy and confidentiality as context-specific and culturally-constructed [71]<br>Processes<br> • RECs could tolerate a lower level of anonymity in the context of health services research–or none where participants want to be identified—than in the case of biomedical research [59]<br> • Creation of safe spaces for disclosure seems necessary across all settings alongside culturally valid criteria for disclosing confidential information and referral/protection systems [69]<br> • Consent forms might need to distinguish between confidentiality and anonymity [30] |
| 5 –Role and competence of researchers | Principles<br>Processes | Principles<br> • Ethics oversight should consider the quality and skill of the researcher(s) as a further factor shaping the level of tolerance acceptable for a research protocol [59]<br> • Researchers' should be responsible for fostering the autonomy of vulnerable participants [65]<br> • Research ethics guidelines should acknowledge that the role of the researcher is inter-relational and constructed continuously on the spot [34], RECs should question the researchers' skills and experience in attending to ethical relationships with their participants [68,70]<br>Processes<br> • Encouraging RECs to ask for a terms of reference document from research teams to outline in advance roles and responsibilities within a participatory study, and anticipate how conflicts will be addressed [62]<br> • Research teams to practise self-reflection across all research stages [72,75] |
| 6 –The working of RECs | PrinciplesProcesses<br> • Increasing demands upon RECs perceived as more related to risk management and compliance concerns [63]<br> • Committees' unfamiliarity with innovative collaborative research approaches [64,74]<br> • Tension between tiers of ethics approval [64] | Principles<br> • Need for a more collaborative, relational approach of research ethics [28,29,56,63,77]<br> • Reframing the role of the RECs as a resource for thinking through ethical issues [35,57,62]<br> • Need for time-effective process for accommodating the emergent design of collaborative research [34,71]<br>Processes<br> • RECs should ask questions in relation to partnership ties and power relations, complexity of roles, differences in expectations between researchers and participants [32]<br> • Reframing the REC involvement in facilitating ethical conduct of participatory research throughout the study [55,57,62,68,77]<br> • Periodic auditing and monitoring of REC decisions [29,30,60] |
| 7 –The research protocol | Principles<br> • Iterative, emergent and cumulative nature of collaborative research not well captured in standard protocols [30,64]<br> • Layers of everyday ethics escaping the requirements of protocols [35]<br>Processes<br> • Rational and technocratic role of RECs expected to check study protocols and supporting documentation in advance of a study and then ensure that these are strictly adhered [29]<br> • Research ethics review forms do not explicitly take into account principles of collaborative research [62] | Principles<br> • Calls for flexible protocols and review procedures [28,57], e.g., when variances from the protocol do not constitute a material change in risk to participants [28]<br>Processes<br> • Staged process of approval of collaborative research, incorporating some prudential flexibility on the level of detail required in protocols in advance of REC approval [30,57]<br> • Protocols of study adopting a collaborative approach should focus on the nature of the research relationship [61]<br> • Use of principles and terminology consistent with collaborative research into existing ethics review forms and guidelines [62,74] |

*(Continued)*

**Table 6.** (Continued)

| Theme | Current system | Options for improvement |
|---|---|---|
| 8 –Seeking consent | Principles<br>• Duty to foster autonomy vs requirement for informed consent [65]<br>• Concerns about the risk of conflating vulnerability with specific requirements of informed consent [35]<br>• Collaborative research blur the differences between researchers and research participants, which can make the consent seeking process uncertain [34]<br>Processes<br>• Unintended consequences of seeking consent in writing [28,55,57]<br>• Arbitrariness of age-based criteria to determine competence to consent among young people (vs parental consent) [56] | Principles<br>• Reframing consent to highlight that participants are consenting to actively engage in the process of inquiry [30]<br>• Endorsing a situational approach to consent seeking [34]<br>• Adopting process-based informed consent [34,76]<br>• Framing of consent as a cultural and social concept [72]<br>Processes<br>• Researchers to provide waivers for written consent [28,72]<br>• RECs to focus on how consent will be negotiated and on avenues for non-participation [68]<br>• Replacing informed consent with written agreements on the type of situations in which participants are informants vs co-researchers [32]<br>• Framing consent seeking as an ongoing, dynamic process, with written consent being obtained some weeks before the interview, and verbal consent being ascertained the day before the interview, immediately before the interview commenced, and at stages during the interview [76] |
| 9 –Compliance with legislation | Principles<br>• Tension created by depth vs breadth of regulation and legislation [77]<br>Processes | Principles<br>• Use of regulation/legislation as a overarching framework for decision making rather than for detailed prescriptions [77]<br>• Universities and policy makers to adapt regulations to the current research environment [74]<br>Processes |
| 10—Integrity, quality, and transparency of research | Principles<br>• Questioning of the ethical principle of scientific integrity and research independence in the context of collaborative research [32]<br>Processes | Principles<br>Processes<br>• Role of gatekeepers in ensuring openness and transparency in all parts of the research process [34]<br>• Agreement on responsibilities for different parts of a research project based on qualifications [32] |
| 11—Accessible findings | Principles<br>Processes | Principles<br>Processes<br>• Research ethics system could support the dissemination of findings of collaborative research in different formats for different audiences [57,62] |
| 12 –Benefits from research | Principles<br>Processes | Principles<br>• Research ethics system needs to acknowledge and value the contribution that collaborative research, alongside other research paradigms and approaches, could make to knowledge generation [59]<br>• Expansion of types of benefits associated with research [38,73,76,78]<br>Processes<br>• Research participants to be encouraged to highlight the personal benefits from participation in research [78] |
| Other | Principles<br>Processes | Principles<br>Processes<br>• Training on research ethics [28,29,32,56,57,60,62,74]<br>• RECs membership [28,30,68,74,77] |

relationships across participants, co-researchers, and researchers, and establish how their power differentials will be addressed [61]). They also fail to accommodate its emergent and iterative nature, e.g., when initial research results inform subsequent data gathering and methodological choices [64], and those layers of everyday ethics which are intrinsic to collaborative research [35].

Privacy and confidentiality are further themes which have been covered in the literature. The blanket approach to anonymity as the bedrock of confidentiality and privacy was disputed: whilst for some research participants this may be appropriate (e.g., in small connected communities [58,69]), for others, protection of anonymity should be flexed to reflect participants' preferences (e.g., in the case of young participants, who may perceive the requirement of anonymity as unjust and discriminatory when based on age [78]). Researchers argued that concepts of privacy and confidentiality are context-specific and culturally-constructed, and research ethics practices should be tailored accordingly.

The literature review highlighted two themes not included in the initial thematic framework. The first was the membership of RECs [28,29,32,56,57,60,62,74]: a recurrent perception was that experience and expertise of qualitative and collaborative research among REC members is minimal, affecting how applications are reviewed and approved. To overcome this, some authors recommend establishing special RECs for collaborative research.

Training was a further additional theme. The literature highlighted the need for REC members to receive training in the full range of research methods and study designs and also recommended that researchers, co-researchers, and gatekeepers engaged in collaborative research should be offered an opportunity to gain a deeper understanding of research ethics, REC culture, and processes [29,32,56,57,60,62,74].

## Exploratory focus group

The analysis of the focus group data identified 9 of the twelve themes of the analytical framework (Table 7).

At a general level, participants described obtaining ethics approval as a bureaucratic hurdle, particularly burdensome for research involving groups deemed vulnerable by research ethics committees

*"I feel the burden of newly found bureaucracy. As we were going through the procedures to prepare for our project, that involved people with dementia, one of the questions that emerged was how much more paperwork will it entail to involve them. Maybe it's a bit trite, but there was a consideration of how can we involve this group less, so the burden on getting ethics is smaller." (FG1-4)*

They also talked about the poor fit between their research and the underpinnings, practices, and language of the research ethics framework within which they were expected to operate. Participants highlighted how the research ethics approval process posed great emphasis on the approval phase of a study, when the study is appraised prospectively, with less relevance given to the actual conduct of research

*"One of the things that I found problematic is the extent to which the whole process is front-loaded. Ethics committees will be very paternalistic in the initial stages of giving approval to a piece of research, but then don't seem to particularly be interested in monitoring the conduct of the research." (FG1-3)*

In relation to Theme 2—Patient, service user, and public involvement, participants highlighted that there could be scope to engage the public and ask what ethical research means to them

*"Some of those concepts [about research ethics] haven't really been aired, in terms of public contribution and what the public would say are important concepts when we think about*

**Table 7. Results of the focus group, by theme.**

| Theme | Current system | Options for improvement |
|---|---|---|
| 1—General ethical principles | Principle<br>• Research ethics system dominated by a biomedical framework<br>• Ethics as bureaucracy in action<br>• The framing of vulnerability<br>Process | Principle<br>• Reframing ethics as a process<br>Process<br>• Establishing a space to discuss ethics with peers<br>• Research ethics mentoring throughout the research study |
| 2 –Patient, service user, and public involvement | Principle<br>Process | Principle<br>• Consulting the public about research ethics<br>Process |
| 3 –Protection of research participants | Principle<br>• Paternalism and over-protection<br>• Vulnerable groups<br>• Unbalanced assessment of risks vs benefits associated with participation in research<br>Process<br>• No processes to highlight positive effects of research for participants | Principle<br>• Shifting towards risk-based approaches to research ethics<br>Process<br>• More acceptable and feasible processes to protect participants |
| 4 –Privacy and confidentiality | Principle<br>Process | PrincipleProcess |
| 5 –Role and competence of researchers | Principle<br>• Lack of trust<br>• Ethos of the researcher<br>Process<br>• Trust in the written word | Principle<br>• Empowering the researcher<br>• Tending towards situated, relational ethics<br>Process |
| 6 –The working of RECs | Principle<br>• Confrontational dynamic between RECs and researchers<br>• Inconsistencies in the way RECs work<br>Process<br>• Front-loaded process<br>• Burdensome process<br>• Lack of monitoring of approved research ethics application | Principle<br>• Relational ways of handling the process by RECs<br>Process<br>• Specialist RECs |
| 7 –The research protocol | Principle<br>• Research ethics system dominated by a biomedical framework<br>Process<br>• Research ethics documents and forms unfit for purpose | Principle<br>• Tending towards situated, relational ethics<br>Process<br>• Simplified research ethics documents and forms<br>• Ethics log |
| 8 –Seeking consent | Principle<br>• One-off, static consent<br>Process<br>• Over-reliance on the written<br>• Mental Capacity Act and consent seeking<br>• Secondary consent, role of consultees | Principle<br>• Reframing consent as an ongoing process<br>Process<br>• Simplifying consent seeking procedures<br>• Personalising the consent seeking process |
| 9 –Compliance with legislation | Principle<br> • Ensuring compliance with the Mental Capacity Act and Data Protection Act<br>Process | Principle<br>Process<br> • Establishing consensus on how to work within the limitations of current legislation |
| 10 –Integrity, quality, and transparency of research | Principle<br>• Regulatory role of research ethics in ensuring integrity of research<br>Process | Principle<br>Process |
| 11 –Accessible findings | Principle<br>Process | Principle<br>Process |

*(Continued)*

**Table 7.** (Continued)

| Theme | Current system | Options for improvement |
|---|---|---|
| 12 – Benefits from research | Principle<br>Process | Principle<br>Process |
| Other -<br>Local research governance arrangements | Principle<br>• Local R&D system dominated by a biomedical framework<br>Process<br>• Fragmented arrangements | Principle<br>Process |

*ethics. There's something about unravelling some of those concepts and saying, are they the right ones actually now? Are they fit for purpose? (. . .) And as soon as you start talking about the public, their view of what is ethical conduct around involvement or around research can feel different.*" (FG1-6)

In relation to Theme 3—Protection of research participants, the group highlighted the protectionist approach that RECs seem to take towards research participants. They underscored the lack of a proportionality in the way risks and benefits are assessed, at times catastrophising the worst scenarios and discounting (or ignoring) possible benefits of research participation

"*What's interesting about it is (. . .) the very high level of paternalism that is triggered when you're trying to organize consent process in a research context (. . .) which sits rather awkwardly with the idea of empowering members of the public, patients, participants in research, to make their own judgments about their involvement. And here we're really only talking about studies which involve interviews and questionnaires from highly experienced researchers.*" (FG1-5)

In their view, this approach was inconsistent with the ongoing discourse around involvement in health and social care research. In particular, they unravelled the tension between the involvement of groups who are seldom heard in research and the perceptions that RECs may have of these groups (such as homeless people as discussed in the extract below)

"*People around the [research ethics] system have a particular view of what homeless people are like, and they're all horribly vulnerable and we should never approach them. And often they're not. They have quite a high degree of agency.*" (FG1-8)

Alongside highlighting the need to shift towards a more proportionate, risk-based approach to research ethics oversight, participants also underscored the need to design processes better suited to ensure participant protection

"*Ethical processes are very important, to make sure that we're protecting the participants, and they are necessary. It's just how we implement them in a way that's most appropriate and acceptable and feasible for the people who are completing them. . .*" (FG1-9)

In relation to Theme 5—Role and competence of researchers, in the experience of participants, the research ethics system seemed to be concerned with protecting participants from possible unethical behaviour of researchers, whose ethos and competencies are undervalued

*"I feel sometimes as if I'm a delinquent, who hasn't yet been found out. The system is geared to treat me as if I'm fundamentally likely to be irresponsible in my dealings with people who are involved in research with me. My experience is that researchers, if anything, are very cautious and careful and thoughtful about the way they engage in this kind of activity."* (FG1-5)

Participants also discussed how the research ethics system is ambivalent in the way the idea of trust towards researchers is framed. On the one hand, the system seems to rely on a front-loaded process that aims to pre-empt ethical issues at the outset of a study and to foresee procedures to mitigate potential issues. On the other, when in the field, researchers operate with substantial degrees of freedom

*"Once you get through, there is quite a lot of trust. All the day-to-day ethical decisions we make as researchers. . . we're being trusted on those."* (FG1-10)

The group agreed that research ethics policies and institutions should move away from research ethics pivoted around the written word and accept the relational and situated dimension of research ethics

*"(Any changes) should be on the basis of trying to introduce more trust into the system. The trustees seem to rely so much on the paperwork rather than on the people with the people, and the trust and responsibilities that people have."* (FG1-8)

Related to Theme 6—The working of RECs, participants described the interactions between research teams and RECs, at times perceived as confrontational and transactional

*"It took three iterations of the REC meeting to be able to get approval in the end to involve people who lacked capacity in the research, and then it was at the expense of having to make other concessions."* (FG1-3)

They also highlighted how the review process of research aiming to involve groups deemed vulnerable was in their experience more burdensome than for other populations

*"We did eventually get approval for that (i.e., a study that included people who lacked capacity). Although it took more times going back to the committee than other projects have."* (FG1-2)

Participants could envision the research ethics approval being handled in a more relational way and as a two-way process, by which RECs and researchers could identify, discuss, and address ethical matters together as the study unfolds. This shift could be beneficial in particular for research with an emergent design, given the expectation that amendments of the research protocol would be required as the research process progresses

*"I wish that the whole process could be more collegiate in the way that it's carried out. That it could be more like a process of mentoring throughout the whole research process, than having to pass an exam at the beginning of it."* (FG1-3)

Participants discussed options to mitigate the uneasiness of some RECs in dealing with collaborative research approaches, specific topics, and research groups that could be perceived as vulnerable

*"I wonder if it's possible to have RECs that are more topic-specific. Or since there are specific consideration for working with vulnerable populations, can there be one or two RECs that specialize on this particular concern? And they apply the principles in a consistent way across the country."* (FG1-1)

Participants reported several issues in relation to Theme 7—The research protocol. The current protocol template was described as unfit for use in the context of collaborative research using emergent designs

*"You find you have to fit your project into a particular kind of language and forms and structure in order to get through."* (FG1-10)

Researchers found the information sheets that they were required to use when recruiting study participants particularly problematic. They felt that these documents looked officious and were hard for the lay public to understand, seemingly protecting the institutions sponsoring the study rather than the participants themselves. Participants suggested that documents supporting participant recruitment could be simplified in many ways

*"Information sheets that it's in one to two pages maximum in decent size font, without all of the sort of legal clauses that are really complicated to understand. (. . .) I'd really like to see it literally in very easy, accessible everyday language."* (FG1-9)

Relatedly, participants also highlighted that in their experience consent-seeking was operationalised in standardised, one-off procedures heavily reliant on the written word (e.g., completing and signing a consent form) rather than as a relational, continuous process. They also raised issues about how the Mental Capacity Act (MCA) is interpreted by RECs. This resulted in limiting the opportunities for individuals with fluctuating capacity and declining cognitive function to take part in research, or in resorting to secondary consent, in contradiction with the discourse around empowerment, or in research lacking real-world relevance

*"During the XX research study, I was put under pressure by a REC to only recruit people who had capacity, even though there would have been a small minority of the people living in that care environment, on the basis that Sections 30–33 of the Mental Capacity Act [which relate to research with people who may lack capacity in England and Wales] say that you can include people who lack capacity only if it's not possible to answer your research question by including people who do have capacity. And I was told by the Chair of that REC that the representativeness of the research was not an issue, as far as they were concerned. It didn't matter if the participants who were actually recruited represented the people living in that care environment or not, because they were going by the letter of the Mental Capacity Act."* (FG1-3)

Lastly, they underscored the burden posed to research by current legislation (or its interpretation by RECs, as illustrated by the extract above on the MCA) and how the legal requirements were embedded in the documentation used for research purposes (e.g., to comply with legislation on privacy and confidentiality) (Theme 9 –Compliance with legislation).

No data were collected in relation to themes 4, 11, and 12. One additional theme on local research governance arrangements when carrying out research involving statutory organisations (e.g., NHS hospital trusts) was identified

*"The local R&D governance processes raise many of the same issues (. . .). The system, certainly at the local level, is still really organized as if everyone's doing a clinical trial" (FG1-5)*

## Discussion

This article provides an overview of the literature on research ethics in the context of collaborative health and social care research and complements it with the perspectives of active academic researchers with experience of navigating the English research ethics system.

The thematic framework that supported the scoping review and the analysis of the focus group data was organised around the normative principles and operational processes identified with the analysis of UK research ethics policies.

In taking this approach, this work focuses on research ethics as a function of the apparatus of research governance (i.e., the broad range of regulations, principles, and standards of good practice in research) and moves away from previous work conflating the discussion of research ethics requirements and ethical practice in research [80–82].

Importantly, using an analytical framework that stems from a normative standpoint allowed comparison of how the research ethics system is *expected* to work vs how it works in practice, according to the literature and based on actual experiences of researchers in the field. We applied the framework to the literature identified with the scoping review and updated and improved on a previous review assessing the literature published between 1990 and 2002 against the ethical principles outlined in the Belmont Report (i.e., autonomy, beneficence, and justice) [83]. We then used the framework to analyse focus group data, generating original empirical evidence about how the research ethics system is currently experienced by active researchers undertaking collaborative research in England.

Our review shows that the literature on this topic is heterogenous (e.g., in terms of language used to identify research using collaborative approaches) and broad in scope. By mapping out how key ethical principles and processes have been dealt with in the literature, it shows that the coverage received by each theme varies substantially. Despite being among the stated principles underpinning research ethics policies, themes around compliance with legislation, integrity, quality and transparency of research, and accessibility of research findings were sparsely covered in the literature. Similarly, they were not discussed by focus group participants as extensively as other topics. Reasons for this are unclear, but it may be that these themes are perceived as peripheral to research ethics per se and are expected to be addressed at different points in the wider research systems (e.g., accessibility of research findings may be addressed prospectively at the point of research funding application and/or at the end of a study, and may be largely seen as a concern of research teams and research funders, not of RECs).

Other themes, such as the working of RECs, the research protocol, and the protection of research participants were found to have received substantial attention in the literature and resonated among focus group participants.

Through the review process, it also became apparent that two themes ('Training on research ethics' and 'REC membership') discussed in the literature were not captured by the analytical framework we developed from policy documents. The lack of consideration of these two themes seems to reflect the dominant epistemological framework underpinning research ethics policies, at least in the UK context, since international guidance addresses both [4]. In relation to training, the UK Policy Framework looks exclusively at competencies and qualifications of research teams (Principle 2 Competence [7]), echoing one of the recommendations of the Helsinki Declaration ("Medical research involving human subjects must be conducted only by individuals with the appropriate ethics and scientific education, training and

qualifications" [3]), taking for granted that REC members are competent to review any type of study regardless of their actual research expertise. Policies do not address REC membership and their expertise either. Historically, this has led to co-option of members with quantitative expertise, well-aligned to the dominant epistemological framework of research ethics, with other types of expertise under-represented [30]. This issue was also pinpointed by focus group participants who aired the idea of establishing RECs with specialist interest in specific research approaches, topics, or populations.

We note that our review did not find effectiveness of research ethics as an emergent theme across the body of the literature we included. Therefore, this work does not contribute to the ongoing debate about the effectiveness of ethics processes [8,9,11,12].

Looking at the results across themes, principles, and processes, findings from the literature and from the focus group were highly concordant. The pattern that emerges from both the literature and the focus group converges around *issues* in how the research ethics system deals with collaborative research and how the research ethics system negatively affects what collaborative research is conducted, and how. On the one hand, this could represent compelling evidence that the positivist underpinnings of research ethics oversight make it unfit for the purpose of reviewing research which does not sit within a positivist paradigm. On the other hand, some of these issues have also been reported in relation to biomedical, experimental research, for example in relation to informed consent [84], risk-benefit assessments [85], and the emphasis on procedures and documents to the detriment of day-to-day conduct of research [86]. Altogether these findings provide empirical corroboration of the concept of ethics creep [63]: the regulatory structure of the ethics bureaucracy has been expanding outward (e.g., taking over research using collaborative, non-experimental approaches in the social sciences) while at the same time intensifying the regulation of practices deemed to fall within its original scope (i.e., biomedical, experimental research). Consistently with this finding, this work adds to the literature documenting burdens associated with research ethics processes [9,87].

Indeed, the overarching discourse that emerges from both the literature and the focus group points towards the need for an overhaul of the rule-based, procedural approach to ethics threaded throughout current regulatory policies and structures, and endorses processes of 'micro ethics' [42], 'situated ethics' [34,88,89], relational ethics [90,91], that place emphasis on 'ethical mindfulness' [92] and reflexivity [93] on the part of individual researchers.

In this vein, this work also offers a rich catalogue of options for improving how the research ethics system could deal with collaborative research. Improvements expressed at the level of principles (e.g., framing consent as an ongoing process or embedding some degree of tolerance around risk, both discussed in the literature and among focus group participants) could open a line of work for research ethics institutions interested in exploring how to translate these principles into institutional processes. Among those expressed at the level of processes, some could be considered for implementation by research ethics institutions (e.g., the creation of specialist RECs, as suggested by focus group participants), others could be used as practical recommendations and advice for researchers navigating the research ethics system.

It is important to recognise strengths and weaknesses of this work. Our analytical framework was based on the analysis of research ethics policies published by UK institutions. However, the themes themselves are consistent with international research ethics frameworks currently in use and the analytical categories of 'Principles' and 'Processes' are broad enough to ensure that the results of the review are relevant beyond the UK context.

To our knowledge, the scoping review is the first attempt to map the academic literature discussing how the research ethics system deals with collaborative research in the health and social care field. The review identified journal articles published between 2010 and 2022 and

indexed in three online academic research databases. Most of the included article were published between 2010 and 2015, when collaborative research approaches became increasingly popular. The academic debate about research ethics flourished at this time [29], with researchers sharing their experience of navigating the research ethics approval process for collaborative studies and often offering guidance and advice to others with similar research interests. This may have contributed developing a community of practice comfortable with addressing the requirements of research ethics. Perhaps as a result, the scholarly interest in writing about this topic in academic outlets subsequently faded away, to be replaced by methodological and good practice guidance published in the grey literature. Although this was outside the scope of this work, we are aware that resources have been made available by organisations such as those under the WHO umbrella (e.g., [94]), national research infrastructure (e.g., [95,96]), University-based Research Ethics Committees, and organisations acting as research facilitators and gatekeepers (e.g., [97]).

The identification of focus group participants began from our professional networks, which may have led to a dominance of participants from certain fields (e.g., dementia care and social care research) and exclusion of others. This could have biased the nature of the discussion towards certain themes or experiences. Also, the focus group was conducted at the beginning of the COVID pandemic, which affected the recruitment of participants.

## Conclusions

The need for regulation and ethics oversight of research using collaborative approaches is not questioned. It is clear, however, that the biomedical regulatory framework currently in use presents obstacles to this type of research. This work documents the wide range of issues that researchers may experience when navigating the research ethics system in relation to research adopting a collaborative approach but, more importantly, it also offers options that could help address those issues within the current framework of research ethics.

It would be unrealistic to expect that every option identified could be immediately taken up and seamlessly implemented, and will then deliver positive results. Some practical recommendations could be voluntarily adopted by research teams and RECs alike and could help streamline some elements or procedures of the research ethics system. However, other options would require formal and procedural changes in research ethics processes that should be initiated by relevant institutions. Some of these may need to be accompanied by fundamental changes in the culture that surrounds research ethics, from it being a bureaucratic, prospective, front-loaded process taking place in a confrontational environment to an opportunity to think through ethical issues throughout a study in the context of a formative and collaborative process.

We hope that this work will help move the debate onwards and contribute to an agenda for change of research ethics for collaborative research in the health and social care field, and beyond.

## Supporting information

**S1 File. Scoping review of the literature–search strings.**
(DOCX)

**S2 File. Preferred Reporting Items for Systematic reviews and Meta-Analyses extension for Scoping Reviews (PRISMA-ScR) checklist.**
(DOCX)

## Author Contributions

**Conceptualization:** Chiara De Poli, Jan Oyebode.

**Data curation:** Chiara De Poli.

**Formal analysis:** Chiara De Poli.

**Methodology:** Chiara De Poli, Jan Oyebode.

**Supervision:** Jan Oyebode.

**Validation:** Jan Oyebode.

**Writing – original draft:** Chiara De Poli.

**Writing – review & editing:** Chiara De Poli, Jan Oyebode.

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
