## [Decision Letter · Decision Letter 0]

10 Oct 2023

PONE-D-23-23653Research ethics and collaborative research in health and social care: analysis of UK research ethics policies, scoping review of the literature and focus group studyPLOS ONE

Dear Dr. De Poli,

Thank you for submitting your manuscript to PLOS ONE. After careful consideration, we feel that it has merit but does not fully meet PLOS ONE’s publication criteria as it currently stands. Therefore, we invite you to submit a revised version of the manuscript that addresses the points raised during the review process. Please take into account the reviewer’s comments and suggestions to improve your manuscript. Please submit your revised manuscript by Nov 24 2023 11:59PM. If you will need more time than this to complete your revisions, please reply to this message or contact the journal office at plosone@plos.org. Please include the following items when submitting your revised manuscript:A rebuttal letter that responds to each point raised by the academic editor and reviewer(s). You should upload this letter as a separate file labeled 'Response to Reviewers'.A marked-up copy of your manuscript that highlights changes made to the original version. You should upload this as a separate file labeled 'Revised Manuscript with Track Changes'.An unmarked version of your revised paper without tracked changes. You should upload this as a separate file labeled 'Manuscript'.If applicable, we recommend that you deposit your laboratory protocols in protocols.io to enhance the reproducibility of your results. Protocols.io assigns your protocol its own identifier (DOI) so that it can be cited independently in the future. For instructions see: https://journals.plos.org/plosone/s/submission-guidelines#loc-laboratory-protocols. Additionally, PLOS ONE offers an option for publishing peer-reviewed Lab Protocol articles, which describe protocols hosted on protocols.io. Read more information on sharing protocols at https://plos.org/protocols?utm_medium=editorial-email&utm_source=authorletters&utm_campaign=protocols.

We look forward to receiving your revised manuscript.

Kind regards,

Alberto Molina Pérez, Ph.D.

Academic Editor

PLOS ONE

“Chiara De Poli had research grant funding from the National Institute for Health Research, School of Social Care Research, grant number 106152/CBF/LSECDP-IF14.

Jan Oyebode currently has research grant funding from the National Institute for Health Research, grant number 204266, Social Care for People with Young Onset Dementia.”

Reviewers' comments:

Reviewer's Responses to Questions

**Comments to the Author**

1. Is the manuscript technically sound, and do the data support the conclusions?

Reviewer #1: Partly

Reviewer #2: Yes

2. Has the statistical analysis been performed appropriately and rigorously? 

Reviewer #1: N/A

Reviewer #2: N/A

3. Have the authors made all data underlying the findings in their manuscript fully available?

Reviewer #1: Yes

Reviewer #2: Yes

4. Is the manuscript presented in an intelligible fashion and written in standard English?

Reviewer #1: Yes

Reviewer #2: Yes

5. Review Comments to the Author

Reviewer #1: The study is designed to address whether the current ethics policies, guidelines and processes are fit for purpose in the context of “collaborative research” in health and social care, but from my perspective there are some key issues :

1. The authors have reviewed and analysed only the UK research ethics policies, including for social sciences, and state that “…. the UK system draws on international standards and governance mechanisms established by the Nuremberg Code and the Declaration of Helsinki, shares this bedrock with other countries, and in this sense it is an example of a modern research ethics system.

However, excellent guidance documents in this area not only from UK itself, (https://www.publicengagement.ac.uk/sites/default/files/publication/cbpr_ethics_guide_web_november_2012.pdf ) but also from international organizations ( https://www.who.int/publications/i/item/9789241502948;
https://www.who.int/publications/i/item/9789241505475;
https://ahpsr.who.int/publications/i/item/2019-12-02-ethical-considerations-for-health-policy-and-systems-research and https://equinetafrica.org/sites/default/files/uploads/documents/PAR_Methods_Reader2014_for_web.pdf) already exist but have not been included nor alluded to. The “Methods Reader” on Participatory Action Research in health systems published in 2014 in collaboration with WHO https://equinetafrica.org/sites/default/files/uploads/documents/PAR_Methods_Reader2014_for_web.pdf discusses the ethical challenges in participatory research very thoughtfully (including many of the issues identified by the authors of this paper) and provides a list of ethical principles relevant for this type of research.

The question to be additionally answered perhaps is why the UK HRA has not embedded the existing guidance including international ethics policies and guidance relevant for participatory action research in its own guidance framework?

In any case, it would be relevant to review these additional documents in the paper, and acknowledge the existence of excellent guidance and principles that already exist.

2. The scoping review was designed to respond to specific questions –

• How does the current research ethics system work for collaborative research in health and social care?

• What are the challenges that the current research ethics framework poses to collaborative research in health and social care?

• What options have been discussed in the literature to overcome these challenges?

It was not designed to answer the question – does the application of the current research ethics framework by ethics committees actually ‘protect’ (the interests of) research participants?

Similarly, the FGD was designed to respond to challenges in obtaining ethics approval, and how the approval processes could be better suited to “collaborative” research. The responses of the FGD participants mostly suggest ways that the ethics committees response could have been more tailored to the type of (collaborative) research. The FGD does not respond to the question of whether interventions made by ethics committee actually protect the interests of the research participants.

Thus the introductory paragraph starting line 58, while perhaps relevant in and of itself, and the statement that (lines 633-637) “……. this work does not identify any instance in which research ethics had been acknowledged to have a positive impact on research nor did we find evidence of whether and why research ethics achieves its objectives….” appear inappropriate for this paper and not justified either by the objectives nor the results of the study. I would suggest deleting these sections.

Here I do not deny the existence of ethics creep nor do I deny the absence of evidence for whether ethics committees actually do what they are supposed to do; the absence of evidence however (to us a clichéd term) is not evidence of absence. Nor does it mean that academicians have not tried to measure the actual effectiveness of ethics review (i.e. that the interests of research participants and communities were protected). Some of these challenges have been very thoughtfully described in the recent article by Lynch et al (https://www.sciencedirect.com/science/article/abs/pii/S0277953621009461)

3. I am a little confused why the authors have chosen to use the term “collaborative” research and not the more widely used “participatory” research. Collaborative research is more generally used for research that is done collaboratively with other researchers (researchers from high income countries working collaboratively with researchers from low and middle income countries, researchers from different sectors (health and agriculture for example) working collaboratively etc.). Actually there are research ethics guidelines for collaborative research of this type (cf https://cioms.ch/wp-content/uploads/2016/08/International_Ethical_Guidelines_for_Biomedical_Research_Involving_Human_Subjects.pdf)

4. Much of the literature that is cited is from before 2015 and hopefully much progress has been made in the last 8-10 years. Most of the articles included in the scoping review are a critical reflection on ethical issues. The authors may wish to comment, based on the reflections, if anything has changed in the past decade or not.

5. Minor point – in line 93, the authors have defined justice rather narrowly in terms of who gets the benefits, whereas research ethics guidance talk about fairness in terms of who gets to participate, who bears the brunt of the risks, fairness of research partnerships and so on. Perhaps the authors may wish to define justice more broadly?

6. Cf line 396 – Membership and training of REC members are covered both in the CIOMS and the WHO guidelines.

Reviewer #2: This article is clearly documented and its methodology is robust. The topic is essential and perfectly analyzed. It may have implications for international contexts beyond UK. Several proposals to improve the Ethics process such as specific RECs and training about the different types of research could apply abroad and be rather simple to implement. The point of participants' data protection (GDPR) may be more underlined as a barrier to RECs decisions in some countries (beyond anonymity or consent). It does not seem to be such a difficulty in the UK according to this article.

6. PLOS authors have the option to publish the peer review history of their article (what does this mean?). If published, this will include your full peer review and any attached files.

Reviewer #1: **Yes: **Abha Saxena

Reviewer #2: **Yes: **Dominique POUGHEON BERTRAND

---

## [Author Response · Author response to Decision Letter 0]

23 Nov 2023

We provide our response to reviewers' comments as an attachment to this submission.

---

## [Decision Letter · Decision Letter 1]

8 Dec 2023

Research ethics and collaborative research in health and social care: analysis of UK research ethics policies, scoping review of the literature and focus group study

PONE-D-23-23653R1

Dear Dr. De Poli,

We’re pleased to inform you that your manuscript has been judged scientifically suitable for publication and will be formally accepted for publication once it meets all outstanding technical requirements.

Kind regards,

Alberto Molina Pérez, Ph.D.

Academic Editor

PLOS ONE

Additional Editor Comments (optional):

Reviewers' comments:

Reviewer's Responses to Questions

**Comments to the Author**

1. If the authors have adequately addressed your comments raised in a previous round of review and you feel that this manuscript is now acceptable for publication, you may indicate that here to bypass the “Comments to the Author” section, enter your conflict of interest statement in the “Confidential to Editor” section, and submit your "Accept" recommendation.

Reviewer #1: All comments have been addressed

2. Is the manuscript technically sound, and do the data support the conclusions?

Reviewer #1: Yes

3. Has the statistical analysis been performed appropriately and rigorously? 

Reviewer #1: N/A

4. Have the authors made all data underlying the findings in their manuscript fully available?

Reviewer #1: Yes

5. Is the manuscript presented in an intelligible fashion and written in standard English?

Reviewer #1: Yes

6. Review Comments to the Author

Reviewer #1: I have reviewed the responses of the authors to my comments and am satisfied with their responses. I thank the authors for taking my review seriously and providing a thoughtful response. The revised manuscript, in my opinion, is more balanced.

7. PLOS authors have the option to publish the peer review history of their article (what does this mean?). If published, this will include your full peer review and any attached files.

Reviewer #1: **Yes: **Abha Saxena

---

## [Editor Report · Acceptance letter]

14 Dec 2023

PONE-D-23-23653R1 

PLOS ONE

Dear Dr. De Poli, 

I'm pleased to inform you that your manuscript has been deemed suitable for publication in PLOS ONE. Congratulations! Your manuscript is now being handed over to our production team.

Kind regards, 

on behalf of

Dr. Alberto Molina Pérez 

Academic Editor

PLOS ONE